# Results of Nailfold Videocapillaroscopy in Patients with Pseudoexfoliative Glaucoma

**DOI:** 10.3390/life13040967

**Published:** 2023-04-07

**Authors:** Urszula Łukasik, Joanna Bartosińska, Ewa Kosior-Jarecka, Dominika Wróbel-Dudzińska, Dorota Krasowska, Tomasz Żarnowski

**Affiliations:** 1Department of Diagnostics and Microsurgery of Glaucoma, Medical University of Lublin, 20-079 Lublin, Poland; 2Department of Cosmetology and Aesthetic Medicine, Medical University of Lublin, 20-093 Lublin, Poland; 3Department of Dermatology, Venereology and Pediatric Dermatology, Medical University of Lublin, 20-093 Lublin, Poland

**Keywords:** pseudoexfoliative glaucoma, nailfold videocapillaroscopy, glaucoma, pseudoexfoliation syndrome

## Abstract

The aim of this study was to evaluate the nailfold videocapillaroscopic examination results from patients with pseudoexfoliative glaucoma (XFG) and to assess the relationship between the results of this examination and the patient’s clinical status in the XFG group. Material and Methods: The studied group consisted of 39 Caucasian patients with XFG and 32 patients in a control group. The patients were classified into two subgroups: the hypertensive pseudoexfoliative glaucoma (hXFG) subgroup and the normotensive pseudoexfoliative glaucoma (nXFG) subgroup. The nailfold videocapillaroscopy (NVC) was performed on all participants. The results of each NVC were classified as having a normal or abnormal pattern. Results: There was no statistical difference between the results of an abnormal NVC pattern in the study group vs. the control group (*p* = 0.8773). Microhemorrhages were shown in 30.0% of patients with nXFG vs. the control group (6.25%) (*p* = 0.0520). Microhemorrhages tended to be more frequent in the XFG group (*p* = 0.1221). A prevalent number of tortuous capillaries was observed in hXFG patients with advanced glaucomatous neuropathy. Dilatation in the capillaries and microbleedings were observed in the group of patients with lower IOP values. Tortuosity in the capillaries was significantly more frequent in PEXG patients (XFG vs. control: *p* = 0.0386). No relationships between the results of NVC and age, c/d, BCVA, time of treatment, and visual field defect were found. Conclusions: Specific features of NVC examination differentiate nXFG from hXFG patients. Some capillaroscopic features may correlate with the patient’s clinical status of XFG.

## 1. Introduction

Pseudoexfoliation syndrome (XFS) is a systemic disease, which is age-related and characterized by the deposition of extracellular fibrillar material in many different tissues all over the body [1]. The presence of exfoliation material in the eye induces some intraocular diseases such as glaucoma, cataract and lens subluxation, or iris atrophy [2,3]. Pseudoexfoliative glaucoma (XFG) typically is a hypertensive form with an open angle, in which the exfoliation material disturbs aqueous humor outflow. However, variants of XFG with normal tension have also been described [4].

Glaucoma is a disease of complex pathogenesis with elevated intraocular pressure (IOP) as a main causative factor. Additionally, vascular disorders are thought to be involved. XFS and XFG are frequently reported in association with systemic vascular diseases. Several studies have reported an association between XFS and cardiovascular as well as cerebrovascular morbidity, aorta aneurysm, Alzheimer’s disease, and elevated homocysteine level in plasma [5,6,7,8]. The pathogenesis of vascular dysfunction in XFS is not well understood. Some studies showed its association with insufficient elastogenesis caused by altered expression of lysyl oxidase-like 1 (LOXL1)**,** elevated homocysteine, or systemic vascular dysregulation [9]. Little is known about the status and possible pathology of microcirculation in XFS.

Nailfold videocapillaroscopy (NVC) is now considered one of the best diagnostic imaging techniques to evaluate microcirculation in vivo [10,11]. This technique is currently used in diagnostics of Raynaud’s phenomenon and microvascular abnormalities in many rheumatic diseases, particularly systemic sclerosis and related conditions. It has also been used in the visualization of microvasculature in other non-rheumatic diseases, including diabetes mellitus and arterial hypertension [12,13]. Abnormalities in NVC have also been reported in some ocular disorders including glaucoma [14,15,16,17].

The aim of the study was to evaluate the results of an NVC examination in patients with XFG and to assess the relationship between the results of this examination and the patient’s clinical status in the XFG group.

## 2. Material and Methods

The studied group consisted of 39 Polish Caucasian patients with XFG treated in the Department of Diagnostics and Microsurgery of Glaucoma, Lublin, Poland. Written consent was obtained from all the patients upon their enrolment in this study. This study adhered to the tenets of the Declaration of Helsinki and the design was approved by the local ethical committee at the Medical University of Lublin (approval number KE-0254/27/2015). The inclusion criterion for this study was as follows: XFG in at least one eye. XFG was diagnosed in cases of optic neuropathy with the glaucomatous damage of the optic disc detected during stereoscopic disc examination, accompanied by the characteristic pattern of the visual field (VF) loss or changes in the retinal nerve fiber layer (RNFL) typical for glaucomatous neuropathy observed in optical coherent tomography (OCT) examination, which were present in patients with diagnosed XFS. The XFS was diagnosed with the presence of dandruff-like exfoliative material on the anterior lens capsule and/or in the anterior segment of the eye. In the case of pseudophakic eyes without detected exfoliation material with a slit-lamp examination, the diagnosis was based on medical records. The exclusion criteria for this study were as follows: primary angle closure glaucoma, primary open angle glaucoma, secondary glaucoma different than PEXG (e.g., traumatic glaucoma, pigmentary glaucoma), and juvenile glaucoma.

Before inclusion, the following parameters were assessed: best corrected visual acuity (BCVA) using Snellen charts with decimal scale; slit lamp biomicroscopy with the evaluation of the anterior segment; gonioscopy; IOP measured with Goldman applanation tonometry; and the stereoscopic fundus examination after pupil dilatation with an assessment of the morphology of the optic disc.

To assess the maximal IOP of untreated patients, 3 measurements of IOP were performed for each eye, the mean value was calculated, and the IOP was assessed during office hours. The patients of the XFG group (*n* = 69 eyes) were classified into 2 subgroups based on assessed maximal IOP: the hypertensive pseudoexfoliative glaucoma (hXFG, *n* = 54 eyes) group with peak intraocular pressure (IOP) above 21 mmHg and the normotensive pseudoexfoliative glaucoma (nXFG, *n* = 15 eyes) group with the highest ever measured IOP ≤ 21 mmHg.

To assess the possible relationship between the results of NVC in the XFG patients, a control group was established. The control group consisted of 32 gender and age-matched patients without diagnosed chronic diseases. All patients from the control group had performed an ophthalmic examination with IOP measurement and stereoscopic eye fundus assessment to exclude XFG and other eye diseases (performed by UŁ). Their medical history concerning chronic general diseases was recorded. The exclusion criteria for both groups were as follows: diabetes mellitus, general hypertension, or any connective tissue diseases. At the time of inclusion into the control group, patients were not taking any medications. In all patients, an NVC examination was performed by one experienced specialist (JB) who was unaware of the patient’s status (studied or control group).

In all studied subjects and in the control group, the NVC was carried out using a videocapillaroscopic VideoCap 3.0 (DSMedica) at 200× magnification. The nailfold capillaries of the 2nd to 5th fingers of both hands were checked. The examination was performed according to the standard protocol. All participants were asked to fast, avoid smoking and drinking caffeinated drinks and alcohol, and avoid taking any drugs possibly affecting the circulatory system. They were also instructed not to remove fingernail cuticles within a month prior to the examination. The examination was performed after 15–20 min of acclimatization to the room temperature of 20–23 °C. A drop of cedar oil was placed on each studied nailfolds to enhance visibility.

The following parameters in the NVC examination were assessed: background (normal/pale), architectural capillary arrangement, number of capillaries, capillary dilatation (enlarged capillaries), presence of tortuous capillaries, glomerular capillaries, neoangiogenesis, microbleedings (microhemorrhages), aneurysmal dilatations, and avascular areas.

The following nailfold videocapillaroscopic features were observed:Within normal limits (Figure 1A);With a few tortuous capillaries (Figure 1B);With a prevalent number of tortuous capillaries;With the presence of enlarged capillaries and microhemorrhages (Figure 1C);With features of neoangiogenesis.

On the basis of the aforementioned features (a–e), the NVC results were qualified as a normal or abnormal pattern:Normal pattern (a–c), i.e., within normal limits with or without the presence of tortuous capillaries;Abnormal pattern (d–e), i.e., with features of microangiopathy.

The studied group consisted of 39 Caucasian patients with XFG and 32 patients in the control group. The study group included 66.7% women (*n* = 26) and 33.3% men (*n* = 13). The mean age was 75.1 y.o. The mean age of women was 75.39 y.o, and the mean age of men was 75.25 y.o. The study group included 69 eyes with XFG. The mean maximal IOP was 26.62 mmHg in the study group, the Mean Deviation (MD) was −12.15 dB, the mean BCVA was 0.59, and the mean c/d 0.70.

The hXFG group included 29 patients: 58.6% women (*n* = 17) and 41.4% men (*n* = 12). The mean age of the hXFG group was 74.24 y.o. The mean maximal IOP was 29.89 mmHg, the MD was −11.49 dB, and the mean BCVA was 0.61.

The nXFG group included 10 patients: 80.0% women (*n* = 8) and 20.0% men (*n* = 2). The mean age of the nXFG group was 78.39 y.o. The mean maximal IOP was 18.66 mmHg, the Mean Deviation was −13.87 dB, and the mean BCVA was 0.56.

The control group included 32 patients: 65.6% women (*n* = 21) and 34.4% men (*n* = 11). The mean age of the control group was 73.5 y.o. The mean age of women was 73.77 yo, and the mean age of men was 73.25 y.o. The mean maximal IOP was 15.41 mmHg in the control group, the MD was 0.36 ±0.7 dB, the mean BCVA was 0.8, and the mean c/d was 0.9.

The details of the demographic features of the groups are listed in Table 1.

In the study group, VF was within normal limits in 19 eyes (37.3%) and there was advanced glaucomatous loss (MD > −12 dB) in 16 eyes (31.4%), hemifield VF defect in 3 eyes (5.9%), arcuate scotoma 10 eyes (19.6%), nasal step in 2 eyes (3.9%), and central scotoma in 1 eye (1.9%).

Statistical analysis of the data was performed using Statistica 12 software, and *p* ≤ 0.05 was considered statistically significant. The results were reported mainly as percentage values, and a normal distribution was checked using the Shapiro–Wilk Test. For non-normally distributed data, the ANOVA Kruskal–Wallis test with Tukey was applied as a post hoc test. The chi-square test with Yates modification, if needed, was used to evaluate the association between glaucoma risk factors and visual field results. In the case of significance obtained with the chi-square test, logistic regression analysis was performed to obtain *p* and assess odds ratios (OR). Correlations were performed with the Spearman test.

## 3. Results

In the study group, we observed a few tortuous capillaries in four subjects (10.26%), a prevalent number of tortuous loops in seven subjects (17.95%), dilatation of capillaries and microhemorrhages in two subjects (5.13%), and the features of neoangiogenesis in eleven subjects (28.21%). In the control group, we observed a few tortuous capillaries in two subjects (5.89%), a prevalent number of tortuous capillaries in twelve subjects (35.3%), dilatation of capillaries and microbleedings (microhemorrhages) in two subjects (5.89%), the features of neoangiogenesis in ten subjects (29.3%). In both groups, avascular areas were not observed. NVC examination outside normal limits (abnormal pattern) with the features of microangiopathy was observed in 33.33% of the study group and 35.19% of the control group (*p* = 0.8773). The details of the NCV results are put in Table 2.

Abnormalities in the arrangement of capillaries (architectural derangement) were observed in 37.93% of patients in the hXFG group, 30.06% of patients with nXFG, 33.3% of the XFG group, and in 9.38% of the control group. Differences in the arrangement in the XFG group and control group were statistically significant (*p* = 0.0332). In the nXFG group, we observed a tendency to a decreased number of capillaries (hXFG vs. nXFG, *p* = 0.0639). A decreased number of capillaries was observed in 10.0% of nXFG patients, whereas it was not observed in the hXFG group or in the control group (*p* = 0.0297).

Dilatation in the capillaries was observed in 37.5% of patients in the control group, 30.0% of patients in the nXFG group, 27.59% of patients in the hXFG group, and 28.21% of patients in the whole XFG group. Tortuosity in the capillaries upon NVC examination was observed in 53.13% of the control group, 44.83% of the XFG group, 41.03% of the total XFG group, and 30.0% of the nXFG group. Tortuous capillaries were significantly less commonly observed in the XFG group than in the control group (*p* = 0.0386). Tortuosity in the capillaries was significantly less frequent in nXFG patients compared to the control group (*p* = 0.0171). Aneurysmal dilatations in the capillaries were observed in 20.0% of patients in the nXFG group, 10.34% of patients in the hXFG group, 12.82% of patients in the XFG group, and 12.5% of patients in the control group. Microhaemorrhages were present in 30.0% of patients with nXFG, in 17.95% of patients with XFG, 13.79% of patients with hXFG, and 6.25% of patients in the control group (nXFG vs. control, *p* = 0.052). The details of the NCV results are put in Table 3.

No relationship between the NVC pattern and maximal IOP was observed. No correlation between the results of the NVC examination and c/d (*p* = 0.8233), age (*p* = 0.3133), time of treatment (*p* = 0.424), type of visual field defect (*p* = 0.7394), or BCVA (*p* = 0.1751) was found. No relationship between the NVC pattern and MD (*p* = 0.4169) and IOP (*p* = 0.8394) was observed.

However, the highest maximal IOP (30.1 ± 17 mmHg) and the most advanced VF defect (MD: −16.4 ± 13.6) were observed in the group of patients with a prevalent number of tortuous capillaries. On the other hand, the lowest maximal IOP (−16.3 ± 5.8 mmHg) was observed in the group of patients with dilatation of capillaries and microhemorrhages.

## 4. Discussion

XFS is an environmentally and genetically co-determined systemic condition, which may involve the tissues all over the body in a clinically significant way [18,19,20] with changes in antioxidant levels in aqueous humor [21]. Glaucomatous neuropathy occurs 6 to 10 times more frequently in eyes with XFS [22,23]. Exfoliation material associated with blood vessels is not always evident clinically since many abnormalities in ocular and systemic vasculature are present in XFS/XFG. Fluorescein angiography examination shows hypoperfusion, fluorescein leakage, visible microneovascularization, and vascular anastomosis of the iris vessels [24,25,26,27,28,29]. Although the causative mechanisms are not fully understood, both XFS and XFG are associated with a dropout of iris capillaries, retinal vein occlusion and neovascularization, cardiovascular dysfunctions, and serious systemic vascular diseases [20,30,31,32]. Exfoliation material has been detected in the myocardium, vessel wall, skin, smooth and striated muscle cells, and many different visceral organs [30,33,34]. In vessels, exfoliation material also changes vascular endothelial cells and contractile pericytes, which are the source of blood vessel wall flexibility and strength. The mentioned alterations in vessels affected by XFS may explain many ocular and systemic vasculature changes, including central retinal vein occlusion [35], vascular ischemia of the iris [36], cerebrovascular and cardiovascular disorders, renal artery stenosis, and aortic aneurysm [37,38,39]. However, there are not many studies evaluating microcapillaries in XFS/XFG patients [9,40,41]. This study showed abnormalities in the arrangement of capillaries to be more frequent in XFS patients.

Mechanisms of vascular dysfunction in XFS are very complex. Homocysteine (Hcy) may affect the tortuosity and avascularity in nailfold capillaries observed in XFS/XFG. Elevated Hcy was previously described in endothelial cell dysfunction during atherosclerosis and small vessel disease [42,43]. Hcy is a non-protein amino acid that is distinct from its homolog, cysteine, because of an additional CH2 bridging group [44]. Hcy was previously reported to be higher in the aqueous humor, tears, and serum of XFS patients [45,46].

Moreover, insufficient elastogenesis caused by disturbed expression of lysyl oxidase-like 1 (LOXL1) [47] involved in the pathogenesis of XFS/XFG [48,49] may result in microvascular changes [9]. Elastic tissue is present in vasculature all over the human body [50]. XFS influences, as an elastotic disorder, the extracellular matrix. It has also been connected to an increased incidence of different elastic tissue disorders, such as pelvic organ prolapse and inguinal hernia [51,52]. Vascular dysfunction in XFS is also relevant to systemic vascular dysregulation (diminished reaction in the cutaneous capillary) [53], disturbed regulation of baroreflex [34], decreased carotid artery distensibility and increased stiffness, and impaired heart rate variability [54] and vascular endothelial function [55]. Maximal IOP in XFG tends to be higher than in primary open-angle glaucoma (POAG) and, according to the Collaborative Initial Glaucoma Treatment Study (CIGTS), reaches on average 31.9 mmHg [56]. Elevated IOP in XFG is related to the obstruction of aqueous humor outflow by exfoliation material and pigment deposition as well as increased outflow resistance, mainly at the level of the trabecular meshwork [57]. However, normotensive variants in XFG have been reported [4]. In this study, the XFG patients were classified into two groups: the hXFG group with maximal IOP above 21 mmHg and the nXFG group with IOP ≤ 21 mmHg. Our results showed that some characteristics of the NVC examination differentiate the nXFG from the hXFG group, i.e., dilatation and tortuosity. This may confirm the hypothesis that in some patients, clinically observed XFS may not be related (or only partially related) to glaucoma pathogenesis and that within XFG patients, there is a subgroup (similar to POAG) of normotensive patients with the causative factors of glaucomatous neuropathy different from elevated IOP. In the absence of elevated IOP, patients with nXFG seem to have more in common with NTG patients than with typical XFG, which was also shown in this study. In such cases, XFS seems to be a factor not affecting the course of the disease. The other possibility is that pseudoexfoliative material may act in some way that is harmful to an optic nerve causing direct neuropathy.

Cousins et al. found that nailfold capillary tortuosity is a distinct non-ocular trait observed in XFS/XFG [9]. Moreover, the prevalence of tortuous capillaries in XFG patients with more severe glaucomatous neuropathy was reported [9]. Tortuosity in the capillaries indicates the advancement of capillary dysfunction. Thereby, it may point out that optic nerve damage in XFG is related not only to elevated IOP but also to vascular factors. Cousins et al. speculate that the accumulation of pseudoexfoliative material in the walls of nailfold capillaries causes nontubular vascular lumens, which, as a result, changes the local hemodynamics [9]. Additionally, XFS may induce alterations in the peripheral vessels with elastosis, vascular endothelial dysfunction, and oxidative stress, leading to vascular stiffness, which decreases and limits normal peripheral blood flow [41]. It is possible that similar disturbances may also negatively impact the vascular bed supporting the optic nerve head.

Some researchers showed that microbleedings were less frequent in PEXG patients compared to POAG [9]. However, when dividing the PEXG patients according to maximal IOP, we observed that dilatation in the capillaries and microbleedings were more prominent in the group of patients with lower values of IOP. Additionally, the decreased number of capillaries in NVC tended to occur in nXFG patients. All these features were previously described as typical for NTG patients [58]. Generally, capillary dilatation is the initial sign of damage to small vessel walls. The existence of homogeneously enlarged loops in microcirculation may be preceded and/or associated with irregular enlargement of microvascular loops indicating initial partial damage to the vessel wall [59].

The prevalence of microhemorrhages in nailfold capillaroscopic examination in nXFG is similar to that observed in patients with NTG. One of the main risk factors of NTG is primary vascular dysregulation with Raynaud’s phenomenon, low blood pressure, migraine, and cold extremities as the symptoms [45,53,60,61]. A spectrum of abnormalities in nailfold capillaroscopy in patients with NTG has been reported. In our previous study, microbleedings, enlargement of capillaries, and branching capillaries were more frequently observed in NTG patients compared to age-matched controls [58]. Park et al. found that nailfold bed hemorrhages may be correlated with the presence of disc hemorrhage in NTG and POAG patients [12].

Cousins et al. showed that nailfold hemorrhages, prominent vascular tortuosity, and avascular zones ≥ 200 μm were more frequently observed in XFS/XFG. The authors concluded that a high degree of nailfold capillary tortuosity was a distinct non-ocular trait observed in XFS/XFG compared to both POAG and controls [9]. However, this study did not differentiate XFG subtypes. Bozic et al. reported no significant differences in the capillary diameter; however, their comparison included POAG and NTG patients [62]. Maric et al. observed that in the XFG group, more patients had narrow or partly narrowed capillaries [40].

Bozic et al. found significantly more intensely spiraled capillaries in NTG patients [62]. In our study, abnormal microvascular loops were observed only in patients with nXFG; they were not present either in the hXFG group or the control group.

Abnormal perfusion and disturbances in the elastic tissue forming lamina cribrosa enhance the risk of glaucomatous damage in XFG [63]. Decreased blood flow within the optic nerve head and the peripapillary retina have been described in eyes with clinically visible exfoliation [64]. Additionally, the possible cytotoxic properties of exfoliation material may be involved [65]. Vascular factors seem to play a pivotal role in the pathogenesis of glaucomatous neuropathy, especially in cases where the level of IOP may not be responsible for optical nerve damage [62].

## 5. Conclusions

The NVC pattern in XFS patients differs from healthy gender- and age-matched control. Additionally, a prevalent number of tortuous loops was observed in more advanced stages of glaucomatous neuropathy. The NCV pattern differs between hXFG and nXFG with microhemorrhages more frequently observed in patients with nXFG, similar to NTG patients. Further studies evaluating OCT angiography will enable a comparison of the NCV pattern and retinal and optic nerve head vasculature in XFG patients.

## Figures and Tables

**Figure 1 life-13-00967-f001:**
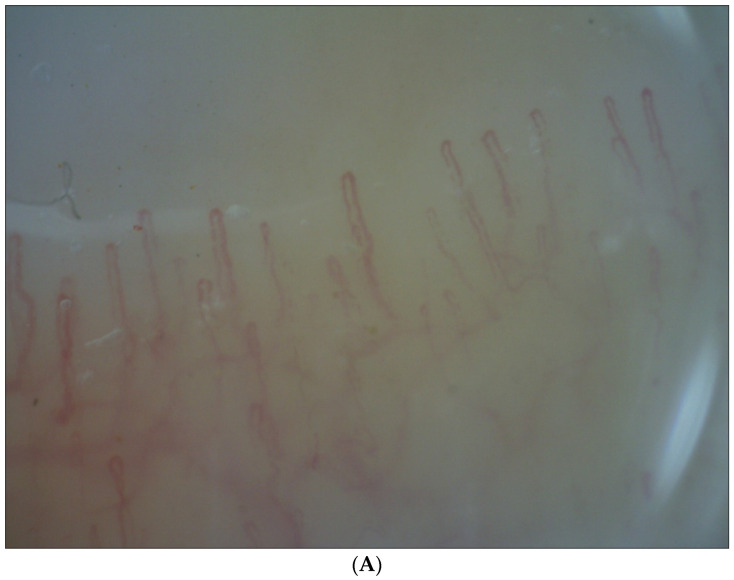
(**A**) Capillary patterns in patients with pseudoexfoliative glaucoma: normal pattern, within normal limits (magnification 200×). (**B**) Capillary patterns in patients with pseudoexfoliative glaucoma: normal pattern, with the presence of tortuous capillaries (magnification 200×). (**C**) Capillary patterns in patients with pseudoexfoliative glaucoma: abnormal pattern, with the presence of enlarged capillaries and microhemorrhages (magnification 200×).

**Table 1 life-13-00967-t001:** Demographic and clinical characteristics of the studied groups.

	XFG (N = 39)	hXFG (N = 29)	nXFG (N = 10)	Control (N = 32)	*p* XFG vs. Control	*p* hXFG vs. nXFG
Gender (Female:Male)	66.7%:33.3%	58.6%:41.4%	80%:20%	65.6%:34.4%	0.8730	0.4048
Age (years)	75.46 ± 7.63	74.24 ± 7.91	78.39 ± 6.40	73.5 ± 5.21	0.748	0.053
Best Corrected Visual Acuity	0.59 ± 0.33	0.61 ± 0.34	0.56 ± 0.33	0.8 ± 0.15	0.3521	0.037
Maximal Intraocular Pressure (mmHg)	26.68 ± 10.74	29.89 ± 11.22	18.66 ± 2.38	15.41 ± 3.82	0.0001 *	0.001 *
Mean Deviation (dB)	−12.15 ± 10.73	−11.49 ± 10.9	−13.87 ± 10.74	0.36 ± 0.7	0.001	0.671

XFG, pseudoexfoliative glaucoma; hXFG, hypertensive pseudoexfoliative glaucoma; nXFG, normotensive pseudoexfoliative glaucoma; *, statistically significant.

**Table 2 life-13-00967-t002:** Results of NVC examination in patients with XFG, hXFG, nXFG, and the control group.

	*n* (%)			
NVC Feature	XFG N = 39	hXFG N = 29	nXFG N = 10	Control N = 32
Pale background	7 (17.95)	5 (17.24)	1 (10.00)	4 (12.50)
Architectual derangement	13 (33.33)	11 (37.93)	3 (30.06)	3 (9.38)
Decreased number of capillaries (<7 capillaries per millimeter)	1 (2.56)	0	1 (10.00)	0
Capillary dilatation (up to 35 µm)	11 (28.21)	8 (27.59)	3 (30.00)	12 (37.50)
Tortuous capillaries	16 (41.03)	13 (44.83)	3 (30.00)	17 (53.13)
Glomerular capillaries	1 (2.56)	0	1 (10.00)	1 (3.13)
Neoangiogenesis	12 (30.77)	9 (31.03)	3 (30.00)	10 (31.25)
Microhaemorrhages	7 (17.95)	4 (13.79)	3 (30.00)	2 (6.25)
Aneurysmal dilatations	5 (12.82)	3 (10.34)	2 (20.00)	4 (12.5)

NVC, nailfold videocapillaroscopy; XFG, pseudoexfoliative glaucoma; hXFG, hypertensive pseudoexfoliative glaucoma; nXFG, normotensive pseudoexfoliative glaucoma.

**Table 3 life-13-00967-t003:** Comparison of NVC examination results for patients with XFG, hXFG, nXFG, and the control group.

	*p*			
NVC Feature	XFG vs. Control	nXFG vs. Control	hXFG vs. Control	hXFG vs. nXFG
Pale background	0.4983	0.7502	0.5863	0.9333
Architectural derangement	0.0332 *	0.7358	0.1251	0.7569
Decreased number of capillaries (<7 capillaries per millimeter)	0.3161	0.0297 *	0.8261	0.0639 ^
Capillary dilatation (up to 35 µm)	0.4196	0.8472	0.4852	0.8092
Tortuous capillaries	0.0386 *	0.0171 *	0.3953	0.1722
Glomerular capillaries	0.8456	0.5886	0.6809	0.9097
Neoangiogenesis	0.9029	0.9508	0.8273	0.9670
Microhaemorrhages	0.1221	0.0520 *	0.4927	0.3246
Aneurysmal dilatations	0.9813	0.6664	0.7671	0.6376

NVC, nailfold videocapillaroscopy; XFG, pseudoexfoliative glaucoma; hXFG, hypertensive pseudoexfoliative glaucoma; nXFG, normotensive pseudoexfoliative glaucoma; * statistically significant; ^ statistical tendency.

## Data Availability

The data are available from corresponding author on request.

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
