# Peer review of "Results of Nailfold Videocapillaroscopy in Patients with Pseudoexfoliative Glaucoma"

_life, 2023, doi:10.3390/life13040967_

Round 1
Reviewer 1 Report
The article is good, require minor corrections:
1. Few language corrections are required.
2.Use detail of the abbreviations used.
3. Improve resolution of images.
4. Conclusion should be in detail and should include future prospects. It's too short.
Author Response
RESPONSES TO THE REMARKS OF REVIEWER 1
The authors would like to thank the Reviewer for all the valuable comments. The changes made according to the Reviewer suggestion are marked with green bold.
Comments and Suggestions for Authors
The article is good, require minor corrections:
Few language corrections are required.
The article was reviewed and improved by professional English translator.
Use detail of the abbreviations used.
The introduction and application of abbreviations were checked and corrected according to the Reviewers suggestion.
Improve resolution of images.
The images are now included in the main manuscript. We believe that during the preparation of the final version we will be able to attach the high resolution version of all images.
Conclusion should be in detail and should include future prospects. It's too short.
The conclusion paragraph was corrected according to the Reviewer’s suggestion.
Reviewer 2 Report
The authors first reported relationship of the nailfold capillaroscopic (NVC) examination with the clinical ophthalmic status in patients with pseudoexfoliative glaucoma (XFG). And they found specific features of NVC 25 examination differentiate nXFG from hXFG patients. However, there are some questions should be addressed.
1. Line 23, the word ‘ore’ maybe mean ‘or’, right?
2. Figure 1~3, it would better to add the scale bar in the images.
3. Figure 1~3, it would better to combine these figures as one figure.
4. Table 2 and 3, it would be better to add some figures to confirm that significant differences in architectural derangement, decreased number of capillaries and tortuous dilatation between the XFG or nXFG vs. control.
5. Line 214, the word ‘ae’ maybe mean ‘an’, right?
6. Line 225 ~226, ‘… changes also vascular endothelial cells and contractile pericytes, …’ should be changed because some words may be lost.
7. Line 296, ‘Cousins et al. r showed …’, ‘r’ should be deleted.
Author Response
RESPONSES TO THE REMARKS OF REVIEWER 2
The authors first reported relationship of the nailfold capillaroscopic (NVC) examination with the clinical ophthalmic status in patients with pseudoexfoliative glaucoma (XFG). And they found specific features of NVC 25 examination differentiate nXFG from hXFG patients. However, there are some questions should be addressed.
The authors would like to thank the Reviewer for all the valuable comments. The changes made according to the Reviewer suggestion are marked with blue bold.
Line 23, the word ‘ore’ maybe mean ‘or’, right?
The typing error was corrected.
Figure 1~3, it would better to add the scale bar in the images.
There is no possibility to add the bar scale at after taking a photo, the information that the magnification of the picture is 200times was added to the description.
Figure 1~3, it would better to combine these figures as one figure.
The photographs were combined in a Photo 1 group.
Table 2 and 3, it would be better to add some figures to confirm that significant differences in architectural derangement, decreased number of capillaries and tortuous dilatation between the XFG or nXFG vs. control.
The presented data are qualitative variables and only percentage bars would be available, that is why the authors decided not to double the results and not to present them in any graphs.
Line 214, the word ‘ae’ maybe mean ‘an’, right?
The typing error was corrected.
Line 225 ~226, ‘… changes also vascular endothelial cells and contractile pericytes, …’ should be changed because some words may be lost.
The sentence was corrected.
Line 296, ‘Cousins et al. r showed …’, ‘r’ should be deleted.
The typing error was corrected.
Reviewer 3 Report
The authors describes the findings of pseudoexfoliative glaucoma explored by naifold- videocapillarosocpoy. Two groups of patients are compared: hypertensive pseudoexfoliative glaucoma (hXFG) subgroup and nor- 15 motensive pseudoexfoliative glaucoma (nXFG).
Abstract: OK
Introduction: Short but concluded.
Some minor issues:
Line 32 ,, is a systemic diseses;;
Line 35 like glaucoma
Material and methods:
Statistics seem OK. However, some morphological parameters should be added ( what is the length, angle, and diameter of capillaries in each stage)
Some minor issues:
Line 92 ,, orecorded,, ?
Line 214 xfs in ae,
Probably some correlation with vasculogenic mimicry should be considered in respect to morphogenesis ( this is only o suggestion)
Discussion: Also at XFS mechanistic ad some lines about: ascorbic acid concentration
Malondialdehyde, Increase concentration of 8-iso-prostaglandin F2a.
Conclusion: OK but statements like To our knowledge, this is the first report correlating the nailfold capillaroscopic examination with the clinical ophthalmic status in patients with XFG -should be avoided.
Author Response
RESPONSES TO THE REMARKS OF REVIEWER 3
The authors describes the findings of pseudoexfoliative glaucoma explored by naifold- videocapillarosocpoy. Two groups of patients are compared: hypertensive pseudoexfoliative glaucoma (hXFG) subgroup and normotensive pseudoexfoliative glaucoma (nXFG).
The authors would like to thank the Reviewer for all the valuable comments. The changes made according to the Reviewer suggestion are marked with violet bold.
Abstract: OK
Introduction: Short but concluded.
Some minor issues:
Line 32 ,, is a systemic diseses;; Corrected
The typing error was corrected.
Line 35 like glaucoma
The typing error was corrected.
Material and methods:
Statistics seem OK. However, some morphological parameters should be added (what is the length, angle, and diameter of capillaries in each stage)
All the information about morphologic parameters of the capillaries possible to be obtained during the capillaroscopy were put in Table 2.
Some minor issues:
Line 92 ,, orecorded,, ?
The typing error was corrected.
Line 214 xfs in ae,
The typing error was corrected
Probably some correlation with vasculogenic mimicry should be considered in respect to morphogenesis ( this is only o suggestion)
After searching the PUBMed we found only the connection between vasculogenic mimicry and tumors. At the moment we cannot find the common point between tumor microcirculation model and neither glaucoma nor capillaroscopy.
Discussion: Also at XFS mechanistic ad some lines about: ascorbic acid concentration Malondialdehyde, Increase concentration of 8-iso-prostaglandin F2a.
The information about possible involvement of anterior chamber antioxidant systems in PEXG pathogenesis were added to the discussion paragraph.
Conclusion: OK but statements like To our knowledge, this is the first report correlating the nailfold capillaroscopic examination with the clinical ophthalmic status in patients with XFG -should be avoided.
The conclusion paragraph was corrected according to the Reviewer’s remark.
Round 2
Reviewer 3 Report
The authors have addressed all the comments.